# Evolution and Genetic Differentiation of *Pleurotus tuoliensis* in Xinjiang, China, Based on Population Genomics

**DOI:** 10.3390/jof10070472

**Published:** 2024-07-10

**Authors:** Peisong Jia, Yarmamat Nurziya, Ying Luo, Wenjie Jia, Qi Zhu, Meng Tian, Lei Sun, Bo Zhang, Zhengxiang Qi, Zhenhao Zhao, Yueting Dai, Yongping Fu, Yu Li

**Affiliations:** 1Engineering Research Center of Chinese Ministry of Education for Edible and Medicinal Fungi, Jilin Agricultural University, Changchun 130118, Chinasunlei@jlau.edu.cn (L.S.); zhangbofungi@126.com (B.Z.); qzx7007@126.com (Z.Q.); daiyueting18@163.com (Y.D.); 2Key Laboratory of Integrated Pest Management on Crops in Northwestern Oasis, Ministry of Agriculture and Rural Affairs, Institute of Plant Protection, Xinjiang Academy of Agricultural Sciences, Urumqi 830091, China; nrzy729@163.com (Y.N.); ly19880219@126.com (Y.L.); jwj202407@163.com (W.J.); zhuqi163vip@163.com (Q.Z.); zhaozhenhao0427@163.com (Z.Z.); 3College of Life Sciences, Xinjiang Agricultural University, Urumqi 830091, China; TM652325@163.com

**Keywords:** *Pleurotus tuoliensis*, evolution, genetic differentiation, population genomics

## Abstract

*Pleurotus tuoliensis* is a unique species discovered in Xinjiang, China, which is recognized for its significant edible, medicinal, and economic value. It has been successfully incorporated into industrial production. Controversy has emerged concerning the evolution and environmental adaptability of this species due to inadequate interspecific ecology and molecular data. This study examines the germplasm resources of *P. tuoliensis* in the Xinjiang region. A total of 225 wild and cultivated strains of *P. tuoliensis* were gathered from seven representative regions. Phylogenetic analysis revealed that seven populations were notably segregated into three distinct groups, primarily attributed to environmental factors as the underlying cause for this differentiation. Population historical size data indicate that *P. tuoliensis* underwent two expansion events, one between 2 and 0.9 Mya (Miocene) and the other between 15 and 4 Mya (Early Pleistocene). The ancient climate fluctuations in the Xinjiang region might have contributed to the comparatively modest population size during the Pliocene epoch. Moreover, through the integration of biogeography and ancestral state reconstruction, it was determined that group C of *P. tuoliensis* emerged initially and subsequently dispersed to groups D and B, in that order. Subsequently, group D underwent independent evolution, whereas group B continued to diversify into groups A and EFG. The primary factor influencing this mode of transmission route is related to the geographical conditions and prevailing wind direction of each group. Subsequent research endeavors focused on assessing the domestication adaptability of *P. tuoliensis* to different substrates. It was found that the metabolic processes adapted during the domestication process were mainly related to energy metabolism, DNA repair, and environmental adaptability. Processes adapted to the host adaptability include responses to the host (meiosis, cell cycle, etc.) and stress in the growth environment (cysteine and methionine metabolism, sulfur metabolism, etc.). This study analyzed the systematic evolution and genetic differentiation of *P. tuoliensis* in Xinjiang. The identified loci and genes provide a theoretical basis for the subsequent improvement of germplasm resources and conducting molecular breeding.

## 1. Introduction

*Pleurotus tuoliensis* is a unique species discovered in Xinjiang, China, which is known for its significant medicinal and economic value [1]. Under natural conditions, it parasitizes the root and stem of *Ferula* plants, mainly distributed in arid areas such as Ili, Tacheng, Altay, and Tori in Xinjiang, China; wild strains are very scarce [2]. In 1987, Mou successfully domesticated *P. tuoliensis* using wood chips and ferulic root chips for the first time and began artificial cultivation research on it in the Xinjiang region [3]. Subsequently, many researchers have researched substrate optimization, yield, nutritional composition, resistance, and other aspects [4,5]. As of 2021, the annual yield of *P. tuoliensis* had reached 51,000 tons, establishing itself as a crucial element in China’s production of edible and medicinal mushrooms, with substantial potential for further development. Wild *P. tuoliensis* resources have accumulated abundant excellent genetic variation, such as disease resistance and drought tolerance, during theirs long-term adaptation to a harsh living environment, and they are valuable genetic resources for breeding and quality improvement. However, the population genetics of wild *P. tuoliensis* resources, such as population genetic diversity, population differentiation, and historical evolution, are still unclear, which is a scientific problem to be solved.

The examination of genetics and differentiation within a population holds significant importance for enhancing the comprehension of species evolution, conservation, and utilization [6]. With the development of sequencing technology, genome analysis has emerged as a valuable tool for elucidating population evolution and adaptive mechanisms. This approach has found extensive applications across various organisms, including animals, plants, and fungi [7,8,9]. *P. tuoliensis*, a significant commercially cultivated edible mushroom in the *P. eryngii* complex species, exhibits a high degree of susceptibility to environmental conditions affecting morphological taxonomic traits, similar to the varieties within the same genus. The evolution and genetic differentiation of this species has always been a controversial issue. Numerous studies have been conducted on the phylogenetics analyses of this group based on molecular markers such as ITS and LSU by both domestic and foreign experts [10,11,12]. Up until 2019–2020, Dai and Liu reconstructed the phylogenetic relationships based on single-copy genes at the genomic level. They suggested that *P. tuoliensis* represents an independently evolved species, serving as a valuable reference for our research [13,14]. The internal evolution, domestication, and environmental adaptability of the *P. tuoliensis* species have not been well resolved.

In this study, a total of 225 wild and cultivated strains from the Xinjiang region were gathered. Resequencing and population genetic analyses were conducted using the genome of *P. tuoliensis* previously published by our research team. This study aimed to analyze the genetic diversity and differentiation within the population of *P. tuoliensis* based on population genetic analyses. By integrating biogeography, this study examined the occurrence pattern of *P. tuoliensis* in the Xinjiang region. Furthermore, utilizing selective elimination analysis, this research investigated the molecular mechanisms of domestication and host adaptability of *P. tuoliensis*. This study will lay a solid foundation for understanding evolution, genetic differentiation, and subsequent improvement of germplasm resources of *P. tuoliensis*.

## 2. Materials and Methods

### 2.1. Sample Preparation

A total of 225 strains of *P. tuoliensis* were collected from Shihezi City, Tori County, Yumin County, Emin County, Fuhai County, Fuyun County, and Qinghe County in Xinjiang Province, China (Table 1). Among these, there were 214 wild strains and 11 cultivated strains. After strain isolation, purification, and identification, all strains were stored at the Strains Preservation Center of the Xinjiang Academy of Agricultural Sciences.

All strains were cultured using PDA medium under conditions of 25 °C for 10 days. Then, 100 mg mycelium from each strain was collected separately, and the DNA of each strain was extracted using a Quality DNA Extraction Kit (DP320, Tiangen bio-chemical technology (Beijing) Co., Ltd., Beijing, China). After undergoing quality detection, the sample was used for subsequent resequencing.

### 2.2. Whole Genome Resequencing and Data Analysis

Entrusted OneMore Technology Co., Ltd., Hong Kong, China, performed whole genome resequencing and data analysis on 225 strains of *P. tuoliensis*, using the Illumina X-ten (Novogene Biotechnology Co., Ltd., Tianjin, China) as the sequencing platform, with a library size of 350 bp and a paired-end sequencing strategy of PE150. The BCL format file of offline data was converted to FASTAQ format by CASAVA V1.8 for subsequent analysis. Using SOAPaligner V2.2.1 and BWA V0.7.12 [15,16], high-quality reads from all short fragment libraries were aligned to the genome sequence of *P. tuoliensis*. Single nucleotide polymorphism (SNP) and Insertion–Deletion (InDel) were predicted using GATK V4.2.1.0 [17]. Principal component analysis (PCA) was then performed on the VCF files of the SNPs from 225 samples using the smartpca program in EIGENSOFT V5.0 [18] and Structure was applied to analyze the genetic structure of the population. Assuming a population subgroup K value between 1 and 10, 10,000 iterations were run each time. SNPhylo [19] was used to extract SNPs from homologous regions of different populations and construct ML trees using MEGA V10 [20].

### 2.3. Analysis of Population Effective Historical Size

Based on the resequencing results of 225 wild strains, the BAM files were compared using the SAMtools tool V1.17 [16] to determine the genotype of the individuals. Bases with low sequencing depth (1/3 of the average sequencing depth) or high sequencing depth (twice the average sequencing depth) were masked. The fq2psmcfa tool V0.6.5-r67 [21] was used to convert diploid consistent sequences into the desired input format file. The generation time (g) was set to 1 year and the mutation rate (μ) was set to 0.2 × 10^−8^ [22].

### 2.4. Analysis of LD and Selection Elimination

Wild and cultivated strains, as well as *Ferula lehmannii* and *F. feruloides*, were set up as two populations and r^2^ was selected as the measure of decay of linkage disequilibrium (LD). The r^2^ value represents the degree of statistical and genetic correlation between two loci (0 < r^2^ < 1). The r^2^ values between each SNP were calculated using Haploview V4.2 [23] and subsequent statistical analysis was performed using R V4.2.1. PoPoolation2 V1.2.2 [24] was used to calculate the Fst values of two populations, vcftools [25] was used to calculate the Pi values of the populations, and a combination of Fst and Pi was used to select the regions identified by both methods as the selected regions and to statistically analyze the genes within the selected regions. Blast2 GO V4.1 and R V4.2.1 [26] were then used to perform GO enrichment and KEGG functional enrichment analysis on the candidate genes, respectively.

### 2.5. Ancestral State Reconstruction

RASP 4.2 [27] was used to speculate on the ancestral status of *P. tuoliensis* in seven regions. The host type and collection location were used for annotation. Firstly, a phylogenetic tree of *P. tuoliensis* was constructed using SNP and maximum likelihood methods, and then the tree and annotation files were imported into RASP 4.2 software. Finally, Bayesian Binary MCMC (BBM) analysis was performed to infer ancestral states [28]. All parameter settings were set by default.

## 3. Results

### 3.1. Genomic Variation in the Population of P. tuoliensis

A total of 225 strains from seven counties and cities in Xinjiang were collected and resequenced, generating 2174.39 Mb of raw reads. After filtering adapters and removing low-quality data, a total of 2149.18 Mb of clean reads were obtained, accounting for 98.96% of strains, with a Q30 value exceeding 92.51%. SAMtools V1.17 and SOAPsnp v1.03 were then used to analyze the genomic variation SNP and InDels using the genome of *P. tuoliensis* as a reference. The results showed that the sequencing depth range of each strain was approximately 18–46×, and the mapping rate was 72.26–92.53% (Appendix A). A total of 4,000,084 high-quality SNPs and 530,097 InDels were then detected in 225 strains (Table 2). Among them, 2,147,423 mutation sites were located in the coding region, 520,213 mutation sites were located in the intron region, 535,958 mutation sites were located in the intergenic region, 317,767 were located upstream of genes, and 284,318 were located downstream of genes in SNPs. After annotating these 2,147,423 SNPs located in the coding region, 299,626 were synonymous mutations, 246,312 were synonymous mutations, and 1,595,538 were unknown. About InDel, 17,973 were located in the coding region, 132,615 were located in the intron region, 66,576 were located upstream of the gene, and 54,758 were located downstream of the gene (Appendix A). This variant information provides new genetic resources for the biology and breeding research of *P. tuoliensis*.

### 3.2. Population Structure and Differentiation of P. tuoliensis

To infer the phylogenetic relationships among 214 populations of *P. tuoliensis* from seven counties and cities, a phylogenetic tree was constructed based on whole genome SNP using the maximum likelihood method based on the likelihood function. All seven populations are generally divided into three major groups, namely, group D, group ABC, and group EFG (see Figure 1A). However, it cannot be ignored that the strains in group D are all clustered outside of one branch, while several strains are distributed as mixtures in other groups. For example, the A6, A7, A1, and BC groups are clustered together, while the C2, C3, C40, and D groups are clustered together. Subgroup ABC can be divided into subgroup A and subgroup BC. The strains in group EFG are mixed, indicating a closer genetic relationship among the strains in this group. The results of PCA and Structure (Figure 1B,C) also support the above conclusion. In addition, we also noticed that a strong differentiation occurred in the ancestor relationship between populations, while the composition of ancestor relationships within each group was similar. This indicates independent evolution among the different groups of *P. tuoliensis,* although each group is distributed relatively close within Xinjiang and there is no strong gene migration. It is speculated that the occurrence of this situation may have been influenced by the local natural environment, leading to adaptive evolution among various groups.

### 3.3. Population Genetic Diversity and Differentiation

FST studies on genetic differentiation between populations showed similar differentiation indices in group ABC, which means relatively close genetic distances between these groups (Table 3). Similarly, group EFG has similar differentiation indices. The differentiation index of group D is not similar to that of the other six groups and is grouped separately. The population nucleic acid diversity indicates that groups C and D have the highest diversity, indicating that these two groups have higher genetic diversity. Next are groups AB and EFG. To some extent, this indicates that the differentiation time of groups C and D is the earliest, and then spreads to groups AB and EFG.

LD analysis was conducted on the population of *P. tuoliensis* in seven different counties and cities in the Xinjiang region, indicating significant differences in the correlation coefficients among the seven groups (Figure 2). Among them, groups A and D have high correlation coefficients, indicating that these two groups have a high level of gene recombination events. When combined with the collection sites of two groups, it was found that they are located in Shihezi City and Emin County, both of which are surrounded by mountains on three sides. Thus, we speculate that the terrain caused less genetic exchange between these two groups and other groups, and the recombination within the population resulted in relatively less genetic diversity in these two groups. Special attention should be paid to group D, which is located in a humid climate and is hosted by *Ferula lehmannii*. Except for limited genetic exchange with other groups, long-term natural selection is also an important reason for the decline in genetic diversity of this group.

### 3.4. Effective Population Size of P. tuoliensis

To explore the historical changes in the *P. tuoliensis* population in Xinjiang, we applied the PSMC method to predict the effective population size changes. In general, the effective population size of *P. tuoliensis* in the seven groups showed the same trend (Figure 3). Among them, there were two instances of population expansion in the *P. tuoliensis* group, between 2 and 0.9 and 15 and 4 Mya, corresponding to the Miocene and Early Pleistocene periods, respectively. The relatively warm and humid environment led to an increase in the number of this group. During a period of 2 to 4 million years, the world entered the Pliocene period, which was generally cold and dry. Extreme weather conditions caused a general decline in the number of *P. tuoliensis* populations. However, it cannot be ignored that the D group has been in a downward trend since 4 million years ago, until a slight increase between 0.3 and 0.08 Mya.

### 3.5. The Origin and Historical Reconstruction of P. tuoliensis in Xinjiang

To study the origin and spread of the *P. tuoliensis* population in the Xinjiang region, we reconstructed the ancestral state of the *P. tuoliensis* population from seven different collection sites (Figure 4A). According to root node 428, the support rates for the origin of the *P. tuoliensis* population in Xinjiang from Yumin County (group C), Emin County (group D), and the CD groups are 73.83%, 19.41%, and 4.22%, respectively, indicating that the *P. tuoliensis* population mainly originates from these two regions, with a greater likelihood of originating from Yumin County. Combined with another important node, 427, with a support rate of 99.37%, it is speculated that *P. tuoliensis* in the Xinjiang region originated from Yumin County and then divided into two branches, group ABC (Shihezi City, Tori County, and Yumin County) and group D. Group D generally exhibits an independent evolutionary state. In addition, we found that the proportion of support originating from group B continued to increase between nodes 421 and 417, indicating that after originating from group C, it gradually propagated towards group B. In group ABC, group A also showed independent differentiation, with a support rate of 69.23% for node 290. It is speculated that after group B spread to A, group A also underwent adaptive differentiation to adapt to the local climate. Node 416 belongs to group EFG (Fuhai County, Fuyun County, and Qinghe County), strongly supporting its origin in Tori County, with a support rate of 95.99%. Therefore, we speculate that the transmission pathways of the three main groups of *P. tuoliensis* in the Xinjiang region originated from Tori County (group C) and then spread to group D and group AB, respectively. Among them, group D underwent independent differentiation, while group B further spread to group EFG. In addition, the host types of the D group and other groups are *F. lehmannii* and *F. ferulaeoides*, respectively. Thus, an analysis of the original host type analyzed was also conducted to further resolve its evolution (Figure 4B). The support rates for root node 428 indicate that the support rates for the original hosts of *F. ferulaeoides* and *F. lehmannii* are 81.28% and 14.20%, respectively. Therefore, we speculate that the original host of *P. tuoliensis* in the Xinjiang region should be *F. ferulaeoides*. Due to the gradual adaptation to the local substrate and humid climate conditions after spreading to Emin County (group D), *F. lehmannii* gradually developed adaptive differentiation.

### 3.6. Selection and Elimination Analysis of Wild and Cultivated Populations

To explore the artificial domestication mechanism of *P. tuoliensis*, selection and elimination analysis was conducted based on the FST values of wild populations (A, B, C, D, E, F, and G) and cultivated populations (Z). The genes from the top five candidate regions were used as candidate genes. A total of 253 candidate genes were obtained. After GO functional enrichment analysis, 66 genes were annotated to 93 GO terms, mainly related to ATPase/amino transmembrane transporter (Figure 5C). The cell membrane is closely related to the material transport and osmotic regulation of resistance under adversity stress, and the large amount of enriched functions related to membrane transport are speculated to be related to the stress resistance adaptability of *P. tuoliensis* under wild conditions. KEGG enrichment analysis revealed that a total of 66 genes were enriched in 57 KO terms, mainly related to energy metabolism, DNA repair, and environmental adaptability, including TCA cycle, mismatch repair, DNA replication, ribosome, aminoacyl tRNA biosynthesis, tryptophan metabolism, lysine degradation, search, and crossover metabolism (Figure 5D). In addition, mitophagy is an important mechanism for maintaining cellular homeostasis in response to reactive oxygen species (ROS). It is closely related to environmental adaptability and may play a role in the adaptation of *P. tuoliensis* to different outdoor and indoor growth environments.

### 3.7. Selection and Elimination Analysis of Different Host Types

To investigate the adaptability of *P. tuoliensis* to different hosts, we selected the D group adapted to *F. lehmannii* and the B group adapted to *F. ferulaeoides* for selective and elimination analysis. The genes from the top five candidate regions were used as candidate genes. A total of 545 candidate genes were obtained. After conducting GO functional enrichment analysis, a total of 145 GO terms were annotated, mainly including DNA binding, intrinsic protein transport, type endopeptidase activity, and membrane coat processes (Figure 5A). A total of 78 metabolic pathways were annotated, which are significantly correlated with meiosis and the cell cycle (Figure 5B). Meiosis provides an important material basis for biological variation and plays a crucial role in the adaptability and evolution of organisms to their environment. In addition, we have identified many metabolic pathways related to drought resistance, such as cysteine and methionine metabolism, sulfur metabolism, tyrosine metabolism, aminoacyl tRNA biosynthesis, etc. It is speculated that these pathways are related to the adaptation of *P. tuoliensis* to the drought growth area of *F. ferulaeoides.*

## 4. Discussion

*P. tuoliensis* is an important species to China. It has been subject to commercial cultivation [29]. According to data from the National Bureau of Statistics, the annual production of *P. tuoliensis* in China has reached approximately 0.4 million tons, with the market demand continually increasing. This indicates significant development prospects for the industry. Throughout the world, the wild resources of *P. tuoliensis* are only reported in Xinjiang and Iran. However, reports of the wild resources in Iran are rare; therefore, Xinjiang is the most important distribution area of wild resources of *P. tuoliensis* in the world. However, the phylogenetic and genetic differentiation issues within the population of *P. tuoliensis* have not been thoroughly elucidated, and are a valuable genetic resource for breeding and quality improvement. The objectives of this study are to clarify the population genetics and environmental adaptability of the wild *P. tuoliensis* populations, which is also a scientific problem to be solved in this study. In this study, a total of 225 wild and cultivated strains of *P. tuoliensis* were gathered from seven representative regions in Xinjiang, China, for whole genome resequencing, and a robust phylogenetic analysis was constructed based on population genomics. Our results demonstrated that the seven populations in Xinjiang, China, are generally divided into three major groups: group D, group ABC, and group EFG. Compared with other strains, the host of group D is *F. lehmannii*, whereas the hosts of the other groups are *F. ferulaeoides*. This suggests that the differentiation among these three groups is primarily driven by the differentiation in hosts. Within the ABC group, subgroup BC is situated in close geographical proximity, specifically in Tori County and Yumin County within the Tacheng area. In contrast, subgroup A is located in Shihezi City. In addition, although the landforms of Yumin County, Toli County, and Shihezi County are similar, the humidity in the Shihezi area is very dry due to the influence of the Tianshan Mountains and Altay Mountains. It is speculated that environmental factors have affected the gene exchange among different varieties, leading to adaptive differentiation over time. Group EFG is located in the Altay region, indicating that the differentiation between this group and group ABC is mainly due to geographical factors.

By reconstructing the evolutionary history of three groups and combining relevant meteorological and geological data, this study found that the effective population size of *P. tuoliensis* fluctuates with changes in the Earth’s climate. Based on previous research findings, *P. tuoliensis* was differentiated from other *Pleurotus* species by approximately 21.9 due to climate change caused by the obstruction of the Indian Ocean warm current into Xinjiang by the uplift of the Qinghai Tibet Plateau [13]. This study further analyzed the population size of the differentiated group of *P. tuoliensis* in Xinjiang, China, and found two instances of population expansion between 15 and 4 and then 0.2 and 0.08 Mya. Between 4 and 15 million years ago was in the middle-to-late Miocene and early Pliocene periods. Studies on ancient plants have shown that during the 18–15 Mya period, the vegetation in the Tianshan Mountains was mainly composed of walnut trees, oak trees, etc., while the spore powder count of pine and cypress was relatively low, reflecting a warm and humid climate condition [30]. In addition, the herbivorous animal Atlantoxerus is a species that lives in warm and humid environments. The herbivorous mammals in the Haramagai and Tonguer animal groups, which belong to this group, are mainly of the low crown type, which also reflects a warm and humid climate characteristic [31]. The above research indicates that the climate conditions in Xinjiang were mainly warm and humid during this period. Perhaps, it is precisely due to such favorable conditions that the population of *P. tuoliensis* increased during this period. After entering the Pliocene approximately 5.3 million years ago, the climate became cold and dry [32], and the inability of the *P. tuoliensis* group to resist changes in stress caused a decrease in the number of groups. During the period of 2 to 0.9 Mya, it entered the Pleistocene. Although significant climate change occurred during the Pleistocene, the overall temperature was relatively warm during this period, and the population size also ushered in a slight increase. But in the late Early Pleistocene, the climate began to cool sharply, and extreme cold conditions caused a rapid decline in the population size of *P. tuoliensis*. In addition, unlike the other six groups, group D is a parasitic group that preys on the species of *F. lehmannii*. It is speculated that the increased number of this group may be related to the increased number of *F. lehmannii* populations. However, no relevant research reports have been found, so we will conduct further research in the future.

The study of species evolution plays a crucial role in revealing the origin and developmental history of species, and a better understanding of how to apply and protect the object *P. tuoliensis* is an important component of the *P. eryngii* species complex that has been involved in human life as one of the most widely consumed mushrooms [33]. Although domestic and foreign mycologists have conduct mass work on the systematic development and distribution of *P. tuoliensis* based on small molecule fragments such as ITS and RPB1, Yang et al. selected 40 single copy genes from the genome level to reconstruct the developmental relationship of *Pleurotus* species and applied biogeographical methods to study the origin and evolution of *Pleurotus* species [34]. However, few reports focus on the origin and evolution of the *P. tuoliensis* population based on genomics. In this study, we further analyzed the origin and transmission history between populations of *P. tuoliensis* from the perspective of population genomes. Our results found two key nodes, 428 and 427, with support rates of 73.83% and 99.37%, respectively, for the origin population in Yumin County, Xinjiang. Therefore, we speculate that *P. tuoliensis* in the Xinjiang region most likely originated from Yumin County, but it cannot be ruled out that it also originated from Emin County. After conducting a geological investigation, we found that the central part of the Tacheng Basin in Emin County is higher in the northeast and lower in the southwest. It is surrounded by mountains on three sides and opens towards the west. Moreover, the dominant wind direction in Yumin County is southwest, which has laid the foundation for the westward expansion of the *P. tuoliensis* side of Yumin County. Due to the influence of terrain, the area is surrounded by mountains on the east, west, and north sides. This results in the north wind being unable to effectively promote the outward spread of *P. tuoliensis* in the area, leading to the gradual formation of independent differentiation in the region. The terrain of Tori County is higher in the south and lower in the north, and its proximity to Yumin County allows *P. tuoliensis* originating from Yumin County to spread to the EFG group located in the northeast direction (Fuhai County, Fuyun County, and Qinghe County) more effectively. This is also the reason why the proportion of support originating from Class B groups continues to increase between nodes 421 and 417. The above results indicate that the *P. tuoliensis* group in the Xinjiang region originated from group C and gradually spread to group B, and from group B to group EFG. It cannot be ignored that group A (Shihezi City) also originated from group B. The terrain in this area is higher in the southeast and lower in the northwest, which is conducive to the transmission of group B of F strains located in the northwest area. Due to the fact that group EFG is located northeast of group A and belongs to the arid Gurban Pass Desert, harsh climatic conditions limit communication between the two groups. In addition, the ancestral host analysis suggests that it originated from the *F. ferulaeoides* that adapted to drought, while the *F. lehmannii* that adapted to humid conditions in group D should have gradually adapted to the local humid climate conditions after the strains of group C spread to group D, resulting in adaptive differentiation.

Compared with most other species of the genus *Pleurotus*, *P. tuoliensis* mainly parasitizes the roots of plants in the Umbelliferae family, which is completely different from the growth substrate of other species growing on decayed broad-leaved trees [35]. To study the host adaptability of *P. tuoliensis*, we selected strains parasitizing on *F. ferulaeoides* and *F. lehmannii* for selective elimination analysis and found that they were mainly significantly correlated with meiosis and the cell cycle. The importance of meiosis in biological genetics and evolution is self-evident, as it generates a large number of variations during the sorting process, which provides an important material basis for organisms to adapt to the environment [36,37]. The meiosis identified in this study was significantly selected, which may provide a genetic basis for the adaptation of *P. tuoliensis* to different substrates. And cell cycle has been proven to participate in immunity and stress resistance in various plants. For example, in the model plant *Arabidopsis*, when attacked by viruses or fungi, the body typically alters the cell cycle to induce related factors or signal transduction, such as CYCD3 and calcium ion signaling, to achieve a lower invasion rate [38,39]. In fungi, evidence regarding the cell cycle involved in response to stress has also been reported. For example, in brewing yeast, Hog1 and cdc28 jointly regulate the cell cycle to cope with osmotic stress [40]. In addition to being associated with host interactions, we have also identified many metabolic pathways related to drought resistance, such as cysteine and methionine metabolism [41], sulfur metabolism [42], tyrosine metabolism [14], and aminoacyl tRNA biosynthesis. Compared with *F. lehmannii*, the growing area of *F. ferulaeoides* is very dry, which results in the different phenotypes of *P. tuoliensis* strains that we collected. The strains growing on *F. ferulaeoides* have more cracks. Therefore, the metabolic pathways related to drought resistance should also play an important role in adapting to the growth environments of two different hosts in *P. tuoliensis*. In summary, the process of adapting to *Ferula* plants includes two aspects: resisting the immune response of living hosts and maintaining survival under drought stress.

Finally, we also explored genes related to the adaptability of *P. tuoliensis* under wild and cultivation conditions, mainly related to energy metabolism, DNA repair, and environmental adaptability. The main challenge that organisms face when transitioning from wild to artificially cultivated conditions is the change in their growth environment [43]. Compared to animals, fungi are unable to achieve a series of stress resistance measures such as migration and increased villi. Generating a series of stress responses to adapt to environmental changes and survive may be the main coping strategy [44]. During the growth and development of organisms, environmental stress often causes various types of DNA damage, such as DNA alkylation [45]. In such cases, it is necessary to repair the matrix related to DNA damage in a timely manner to avoid serious consequences, such as loss of genetic information and modification that affects growth and development. The various processes identified in this study, such as DNA damage repair and mismatch repair, may play an important role in maintaining nucleic acid homeostasis in *P. tuoliensis* in different environments. Similarly, amino acid metabolism also plays an important role in response to adversity in wild conditions. For example, tryptophan is a precursor substance for the metabolism and synthesis of indole-3-acetic acid, which can respond to stressors such as drought and heat in plants and edible fungi [46]. Energy metabolism is also an important factor in the process of domestication and adaptation. For example, Sun et al. demonstrated that the energy metabolism domestication response ability of tropical species, such as the Southern Grass Lizard, is significantly lower than that of temperate species, such as the Northern Grass Lizard and the White-Striped Grass Lizard [47]. Starch and sucrose metabolism are the basic units involved in cellular nutrient absorption, catalyzing the hydrolysis of lignocellulose for nutrient absorption. During the transition from outdoor to indoor cultivation, the substrate of *P. tuoliensis* usually changes from Umbelliferae plants to substrate materials such as sawdust and bran. Thus, starch and sucrose metabolism pathways are closely related to the changes in cultivated substrates. In addition, we have annotated many pathways related to ribosomes and transmembrane transporters. Plants undergo significant changes in gene expression under stress and undergo protein synthesis in ribosomes [41]. Therefore, we speculate that these metabolites are closely related to the transport and osmotic regulation of resistant substances and play an important role in the adaptability of *P. tuoliensis* to the wild environment by regulating gene expression. However, the life activities of organisms require multiple mechanisms to coordinate and cooperate to complete, and we will conduct more in-depth research on adaptability and other aspects in the future.

## 5. Conclusions

In this study, we published 225 resequencing data of *P. tuoliensis* and researched its evolution and genetic differentiation using this data. All 225 strains collected from seven counties and cities in the Xinjiang region were generally divided into three groups: group ABC, group D, and group EFG. Through the reconstruction of ancestral states, we discovered that *P. tuoliensis* in Xinjiang originated in Yumin County and subsequently spread to Emin County and Tori County. Among them, strains in Emin County have undergone independent differentiation due to the influence of terrain, while the strains in Tori County have further spread to other areas. The effective size of the seven populations of *P. tuoliensis* indicates two instances of population expansion, ranging from 0.2 to 0.09 and 15 to 4 million years. This expansion was mainly adapted to the ancient climate change in the Xinjiang region. Finally, we conducted intergroup selection elimination analysis to investigate the domestication and host adaptability of the *P. tuoliensis*. The metabolic processes to be adapted during the domestication process are mainly related to energy metabolism, DNA repair, and environmental adaptability. The processes adapted to the host are divided into two parts: one is the response to the host (meiosis, cell cycle, etc.), and the other is the response to the stress of the growth site (cysteine and methionine metabolism, sulfur metabolism, etc.). The genomic data and analysis generated in this study will provide valuable resources for studying the evolution, genetic differentiation, and high-quality breeding of a new variety of *P. tuoliensis*.

## Figures and Tables

**Figure 1 jof-10-00472-f001:**
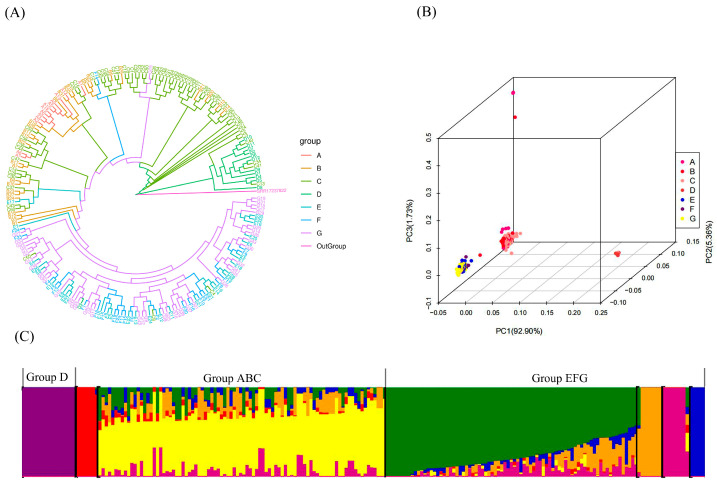
Phylogenetic analysis of the population and genetic diversity of *P. tuoliensis* based on population genetics. (**A**) Phylogenetic tree constructed based on SNP; (**B**) Principal component analysis of clustering relationships among various strains; (**C**) Structure analysis of ancestor relationship among different strains.

**Figure 2 jof-10-00472-f002:**
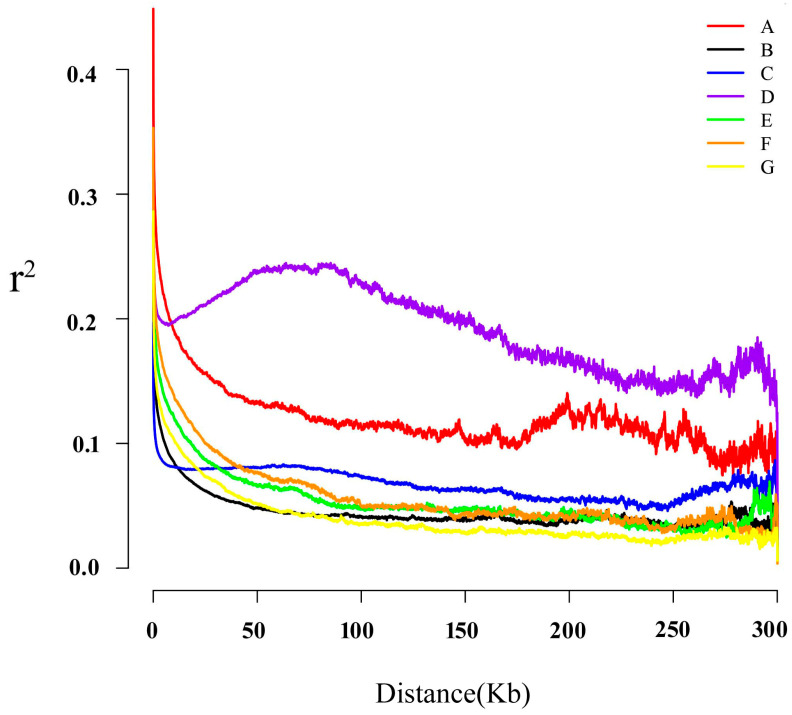
LD analysis in seven groups of *P. tuoliensis*.

**Figure 3 jof-10-00472-f003:**
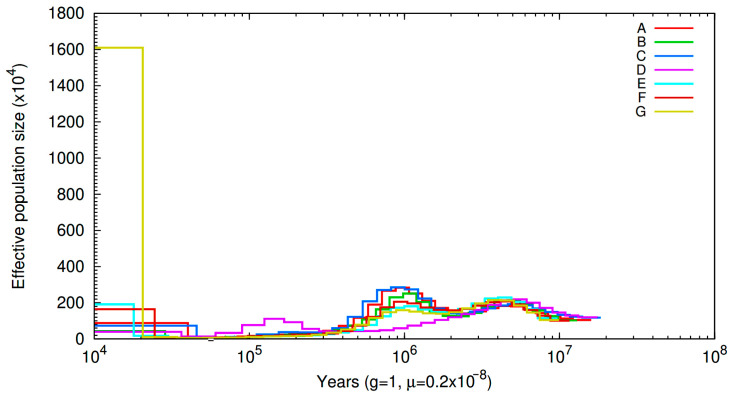
Effective population size in seven groups of *P. tuoliensis*.

**Figure 4 jof-10-00472-f004:**
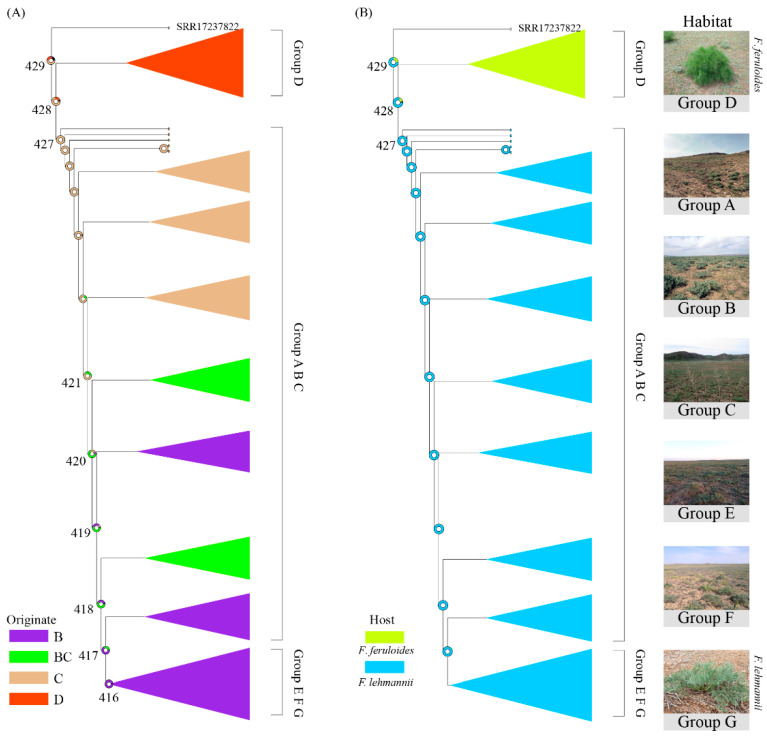
The origin and historical reconstruction of *P. tuoliensis* (**A**) and host type (**B**) in Xinjiang.

**Figure 5 jof-10-00472-f005:**
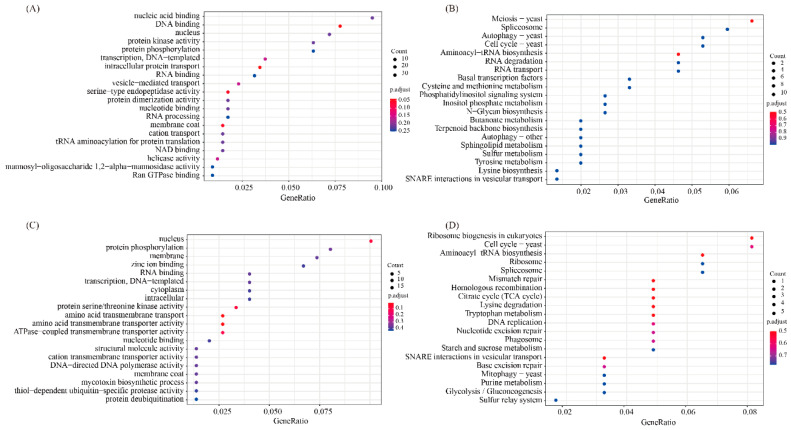
Enrichment analysis of selected functional genes. (**A**) GO enrichment analysis of selected genes in different host types; (**B**) KEGG enrichment analysis of selected genes in different host types; (**C**) GO enrichment analysis of selected genes in domestication cultivation; (**D**) KEGG enrichment analysis of selected genes in domestication cultivation.

**Table 1 jof-10-00472-t001:** Strains of *P. tuoliensis* used for genome resequencing.

Population Type	Origin	Population Code	Number
Wild strains	Shihezi City, Xinjiang, China	A	13
	Tori County, Xinjiang, China	B	29
	Yumin County, Xinjiang, China	C	62
	Emin County, Xinjiang, China	D	12
	Fuhai County, Xinjiang, China	E	31
	Fuyun County, Xinjiang, China	F	24
	Qinghe County, Xinjiang, China	G	43
Cultivated strains	Changchun City, Jilin, China	Z	11
Total			225

**Table 2 jof-10-00472-t002:** SNP and InDel statistics in *P. tuoliensis*.

Type	Number (SNP/InDel)	Percentage (%) (SNP/InDel)
Total	4,000,084/530,097	100/100
Intergenic	535,958/69,079	13.4/13.03
Upstream	317,767/66,576	7.94/12.56
Exonic	2,145,854/160,814	53.65/30.34
Intronic	520,213/132,615	13.01/25.02
Splicing	7303/2495	0.18/0.47
Exonic, splicing	1569/182	0.04/0.03
Upstream, downstream	187,102/43,499	4.68/8.21
Downstream	284,318/54,758	7.11/10.33

**Table 3 jof-10-00472-t003:** Genetic diversity analysis in seven groups of *P. tuoliensis*.

	Fst						Pi
	B	C	D	E	F	G	
A	0.0146 ± 0.0335	0.0170 ± 0.0359	0.3253 ± 0.2573	0.0510 ± 0.0703	0.0447 ± 0.0655	0.0559 ± 0.0781	0.0003 ± 0.0008
B		0.0079 ± 0.0152	0.3188 ± 0.2725	0.0289 ± 0.0464	0.0256 ± 0.0433	0.0335 ± 0.0500	0.0003 ± 0.0008
C			0.2581 ± 0.2366	0.0311 ± 0.0385	0.0273 ± 0.0370	0.0365 ± 0.0435	0.0004 ± 0.0010
D				0.3662 ± 0.2964	0.3600 ± 0.2890	0.3736 ± 0.3039	0.0004 ± 0.0010
E					0.0015 ± 0.0155	0.0039 ± 0.0150	0.0002 ± 0.0005
F						0.0019 ± 0.0148	0.0002 ± 0.0005
G							0.0002 ± 0.0005

## Data Availability

The raw data supporting the conclusions of this article will be made available by the authors on request.

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
