# Peer review of "Evolution and Genetic Differentiation of Pleurotus tuoliensis in Xinjiang, China, Based on Population Genomics"

_jof, 2024, doi:10.3390/jof10070472_

Round 1
Reviewer 1 Report
No hypotheses are tested, and the analyses are not justified. As it is, it is just a lot of analyses done on a data set. Then the discussion becomes very difficult to understand as reader you are expected to find a discussion on the questions raised in the introduction.
not applicable at this state
Author Response
Dear Editors and Reviewers:
Thank you for your letter and the reviewers’ comments regarding our manuscript entitled
Evolution and Genetic Differentiation of Pleurotus tuoliensis in Xinjiang, China Based on Population Genomics (Manuscript ID: jof-3052304). Those comments are all valuable and very helpful for revising and improving our manuscript, as well as providing important guidance for our researches. We have carefully studied the comments and made correction that we will hope meet with approval. The reviewer comments are laid out below in bold and italicized font, and specific concerns have been numbered. Our response is provided in regular font, and modifications/additions to the manuscript are indicated in red text. The main corrections in the paper and the respondes to the reviewers’ comments are as follows:
Responds to the reviewers’ comments:
Reviewer #1:
Comment 1: No hypotheses are tested, and the analyses are not justified. As it is, it is just a lot of analyses done on a data set. Then the discussion becomes very difficult to understand as reader you are expected to find a discussion on the questions raised in the introduction.
Response: (Introduction, Page 2, Lines 55-57; Discussion, Page 9, Lines 303-313) According to Reviewer’s comment, we have carefully studied the comments and made correction that we will hope meet with approval as follow. (1) In this study, the objectives of this study are to clarify the population genetics and environmental adaptability of the wild Pleurotus tuoliensis populations, and which is also a scientific problem to be solved in this study. (2) Whole-genome resequencing is used to population genetics and environmental adaptability of wild P. tuoliensis populations. This technique is now a mainstream tool in population genetics and environmental adaptability research, it can accurately and comprehensively analyze the genetic diversity, genetic differentiation, environmental adaptive evolution and historical evolution of populations. The results of the study were also analyzed based on whole-genome resequencing data, and the results are reasonable and reliable. (3) In the introduction, we also put forward the scientific problems to be solved in this study, witch is to clarify the population genetics and environmental adaptability of the wild P. tuoliensis populations. (4) In the discussion, we further clarified the value of wild P. tuoliensi as a resource, the purpose of this study and the scientific problems to be solved. In addition, the genetic population division, population evolution history, host adaptive evolution and artificial adaptive evolution of wild P. tuoliensis populations were separately discussed in detail.
Comment 2: It is very difficult to see how the authors define the groups they are describing. It seems that it is based on the phylogenetic analysis (fig 1A), but I am no convinced. The only group I can see is group D. The authors will have to explain how they define the other groups. The rationale behind the Structure analysis is unclear. What do the authors want to show, and which parameters were used. The only thing that is mentioned is that 10,000runs were done, which is rather low.
Response: (Introduction, Page 2, Lines 55-57) According to Reviewer’s comment, we have carefully studied the comments and made correction that we will hope meet with approval as follow. (1) The definition of three genetic populations is based on phylogenetic analysis, principal component analysis and genetic structure analysis. By looking at the phylogenetic tree, we can see clearly that group D is a single branch, group ABC is together, group EFG is together. By looking at the PCA diagram, we can more clearly see that group D is a single branch, group ABC is together, group EFG is together. Of course, we can't ignore the fact that there are a few strains that are distributed in a mixture of other strains, but overall, we think it's reasonable to divide into three genetic groups, this is closely related to the geographical location and evolutionary history of the distribution of wild resources. (2) The diagram of the genetic structure has been adjusted to make the grouping of geographic populations look clearer. (3) 10000 run iterations each time for genetic structure analysis is the most common in most studies, including many top journals such as nature and cell.
For the convenience of Reviewers and Editors, we will submit the manuscript with both marked and unmarked versions. We appreciate the Editors/Reviewers’ warm work and hope that the corrections will meet with approval.
Once again, thank you very much for your comments and suggestions.
Sincerely.
Peisong Jia
494744038@qq.com
Reviewer 2 Report
This is an extensive research paper on Pleurotus tuoliense. However, a few additional things are needed.
1. To help readers understand, detailed information on the regional information and cultivated strains of each wild sample is needed. It is recommended that it be added to the text as a map or table.
2. I need detailed information about the kit used in the study. Please provide detailed information such as model and manufacturer rather than the CWBIO Qulaity DNA extraction kit.
3. In the introduction, the value of Pleurotus tuoliense as a resource should be written in more detail to further solidify the validity of this study.
This is an extensive research paper on Pleurotus tuoliense. However, a few additional things are needed.
1. To help readers understand, detailed information on the regional information and cultivated strains of each wild sample is needed. It is recommended that it be added to the text as a map or table.
2. I need detailed information about the kit used in the study. Please provide detailed information such as model and manufacturer rather than the CWBIO Qulaity DNA extraction kit.
3. In the introduction, the value of Pleurotus tuoliense as a resource should be written in more detail to further solidify the validity of this study.
Author Response
Dear Editors and Reviewers:
Thank you for your letter and the reviewers’ comments regarding our manuscript entitled
Evolution and Genetic Differentiation of Pleurotus tuoliensis in Xinjiang, China Based on Population Genomics (Manuscript ID: jof-3052304). Those comments are all valuable and very helpful for revising and improving our manuscript, as well as providing important guidance for our researches. We have carefully studied the comments and made correction that we will hope meet with approval. The reviewer comments are laid out below in bold and italicized font, and specific concerns have been numbered. Our response is provided in regular font, and modifications/additions to the manuscript are indicated in red text. The main corrections in the paper and the respondes to the reviewers’ comments are as follows:
Comment 1: To help readers understand, detailed information on the regional information and cultivated strains of each wild sample is needed. It is recommended that it be added to the text as a map or table.
Response: (Materials and methods, Page 3, Line 98) According to Reviewer’s comment, we have added the table with information on population type, source, population code, and number of wild and cultivated P. tuoliensis strains in materials and methods.
Comment 2: I need detailed information about the kit used in the study. Please provide detailed information such as model and manufacturer rather than the CWBIO Qulaity DNA extraction kit.
Response: (Materials and methods, Page 2, Line 95) According to Reviewer’s comment, we have provided the detailed information of DNA extraction kit such as model and manufacturer in materials and methods.
Comment 3: In the introduction, the value of Pleurotus tuoliense as a resource should be written in more detail to further solidify the validity of this study.
Response: (Introduction, Page 2, Lines 51-57) According to Reviewer’s comment, we have written the value of wild P. tuoliensis as a resource in more detail to further solidify the validity of this study, and they particularly are valuable genetic resource for breeding and quality improvement in the introduction.
For the convenience of Reviewers and Editors, we will submit the manuscript with both marked and unmarked versions. We appreciate the Editors/Reviewers’ warm work and hope that the corrections will meet with approval.
Once again, thank you very much for your comments and suggestions.
Sincerely.
Peisong Jia